# PerturbPFN: Probing the Limits of Synthetic Priors in Perturbation Modelling

Yuche Gao [1]   José Miguel Hernández-Lobato [1]   Siyuan Guo [2 1]

## Abstract

Predicting cellular responses to unseen chemical perturbations is challenging due to unknown mechanisms, high-dimensional responses, and limited experimental coverage. We propose PerturbPFN, a PFN-style amortized model for unknown-target perturbation prediction under a hierarchical synthetic structural prior. Rather than directly regressing expression responses, PerturbPFN infers a latent system graph, sparse intervention targets, and intervention strengths, then propagates their effects through an SCM decoder. The model is trained entirely on prior-predictive synthetic episodes generated from biologically motivated graph and expression simulators, enabling structured in-context inference without test-time gradient updates. We evaluate PerturbPFN on both real single-cell perturbation data and synthetic benchmarks, covering effect prediction, target identification, and regulatory structure discovery. Our results show that PerturbPFN offers a complementary trade-off to specialized baselines, achieving competitive perturbation prediction with low inference cost while exposing interpretable intermediate estimates of targets, strengths, and system structure.

## 1. Introduction

Predicting how complex biological systems respond to novel interventions is a central challenge in drug discovery. The design of clinical therapeutics often relies on chemical perturbations (Sadybekov & Katritch, 2023). Small-molecule drugs introduce a significantly more complex intervention modality than discrete genetic edits, such as CRISPR knock-outs (Maathuis et al., 2010). For example, the perturbation target, pathway, and effective mechanism may be unknown,

context-dependent, or only partially captured by perturbation covariates. In this regime, effect prediction requires not only modelling the post-perturbation distribution, but also inferring latent atomic targets and mechanisms from observed response data and perturbation features (Mooij et al., 2020; Schneider et al., 2025). Further, because the theoretical design space of these small molecules is vastly larger than what can be physically screened in a laboratory, interpolating between known drugs is insufficient. Predictive models must learn to generalize to unseen perturbations (Tejada-Lapuerta et al., 2025).

Existing deep learning approaches for perturbation prediction generally fall into three distinct paradigms. First, unstructured *black-box* models directly learn the statistical distribution shifts induced by a perturbation (Hetzel et al., 2022; Bunne et al., 2023; Lotfollahi et al., 2023), performing inference within a learned latent space. Second, structured *mechanistic* models frame perturbation responses as explicit interventions on an underlying—often causal—representation of the system (Parascandolo et al., 2018; Gonzalez et al., 2025; Roohani et al., 2024), offering advantages in interpretability. Finally, recent foundational model developments seek to learn robust representation spaces by pre-training on massive scales of real-world experimental data, utilizing broad biological priors as seen in scGPT (Cui et al., 2023), GenePT (Chen & Zou, 2023), and LPM (Miladinovic et al., 2025).

However, current structured approaches often rely on instance-specific optimization, limiting their ability to amortize knowledge across heterogeneous contexts such as differing cell lines, experimental batches, or diverse perturbation panels. To overcome these limitations, we draw inspiration from Prior-Data Fitted Networks (PFNs), which have demonstrated that pre-training on large-scale synthetic tasks can amortize the inference process (Müller et al., 2022; Hollmann et al., 2025; Robertson et al., 2025; Qu et al., 2025; Grinsztajn et al., 2025). This enables "in-context" prediction on novel datasets without the need for task-specific gradient updates.

Motivated by this perspective, we propose *PerturbPFN*, a PFN-style model that formulates unknown-target perturbation effect prediction as amortized inference under a hierarchical synthetic SCM prior. The prior generates biologically

[1]University of Cambridge, Cambridge, United Kingdom [2]Prior Labs, Freiburg, Germany. Correspondence to: Yuche Gao <yg473@cam.ac.uk>, Siyuan Guo <sguo26v@gmail.com>.

*Proceedings of the $2^{nd}$ ICML Workshop on Foundation Models for Structured Data*, Seoul, South Korea. 2026. Copyright 2026 by the author(s).

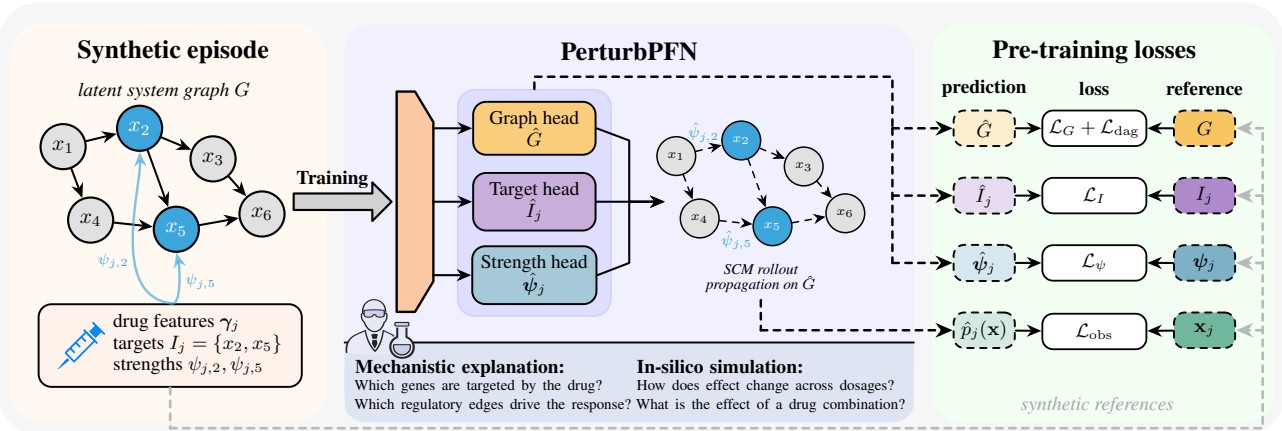

*Figure 1.* PerturbPFN pretraining pipeline. Synthetic episodes provide latent references for the system graph $G$, perturbation targets $I_j$, intervention strengths $\psi_j$, and outcomes $\mathbf{x}_j$. The model predicts a graph and query-specific intervention variables, propagates them through an SCM decoder to predict perturbation effects, and is trained with multi-task losses on all synthetic references.

tailored graphs, perturbation targets, mechanism parameters, and response samples; a transformer-based model learns from these prior-predictive episodes to infer latent quantities and predict held-out perturbation responses. At test time, the model maps context observations and perturbation covariates to estimates of latent structure, targets, and mechanisms, then predicts post-intervention gene-expression distributions through the corresponding intervened generative process. Our contributions are:

- We introduce PerturbPFN, a PFN-style framework trained on biologically grounded synthetic episodes that provide supervision for latent mechanisms typically unavailable in empirical perturbation data.

- We propose a structured intervention bottleneck that predicts sparse atomic targets and low-dimensional intervention strengths, then propagates their effects through an inferred SCM decoder instead of directly predicting high-dimensional responses.

- We evaluate on real single-cell perturbation data from Srivatsan et al. (2020) and synthetic datasets with known latent structure, covering effect prediction, target identification, and structure discovery.

## 2. Methodology

### 2.1. Problem setup.

We study episodic perturbation prediction. Each episode represents one system observed under multiple perturbation regimes, such as a cell line exposed to different drugs. The system contains $p$ measured variables. Each observation is written as $(\boldsymbol{\gamma}_i, \mathbf{x}_i)$, where $\mathbf{x}_i \in \mathbb{R}^p$ is the measured system state and $\boldsymbol{\gamma}_i \in \mathbb{R}^d$ is a regime descriptor that parameterizes experimental conditions. For example, $\boldsymbol{\gamma}_i$ may encode the chemical embedding of a specific treatment, a

continuous scalar for dosage levels, or a categorical indicator for control treatments. Given context observations $\mathcal{D}_{\mathrm{ctx}} = \{(\boldsymbol{\gamma}_i, \mathbf{x}_i)\}_{i=1}^{n_{\mathrm{ctx}}}$, and a query regime $\boldsymbol{\gamma}_j$, the desired prediction is the posterior predictive distribution (PPD)

$$p(\mathbf{x}_j \mid \boldsymbol{\gamma}_j, \mathcal{D}_{\mathrm{ctx}}) = \int p(\mathbf{x}_j \mid \boldsymbol{\gamma}_j, \mathbf{t}) \, p(\mathbf{t} \mid \mathcal{D}_{\mathrm{ctx}}) \, d\mathbf{t},$$

where $\mathbf{t}$ denotes latent task variables, including the system structure and perturbation-response mechanisms that map each regime descriptor $\boldsymbol{\gamma}_j$ to unobserved atomic targets and intervention strengths.

**Latent Structural Causal Model (SCM).** We assume that each episode is governed by a latent causal generative process $\mathcal{M} = \{G, \boldsymbol{\theta}\}$, where $G \in \{0,1\}^{p \times p}$ is the adjacency matrix of a directed acyclic graph (DAG) over the measured variables and $\boldsymbol{\theta}$ parameterizes the local mechanisms. A perturbation regime $\boldsymbol{\gamma}$ does not directly determine the full high-dimensional response. Instead, it induces a latent atomic intervention $\mathcal{I} = \{I, \psi\}$, where $I \in \{0,1\}^p$ denotes sparse intervention targets and $\psi$ denotes their strengths. We factorize this intervention model as

$$p(\mathcal{I} \mid \boldsymbol{\gamma}, \boldsymbol{\phi}) = p\big(I; g_{\boldsymbol{\phi}}(\boldsymbol{\gamma})\big) p\big(\psi; h_{\boldsymbol{\phi}}(I, \boldsymbol{\gamma})\big), \quad (1)$$

where $g_{\boldsymbol{\phi}}$ and $h_{\boldsymbol{\phi}}$ are descriptor-to-intervention maps parameterized by $\boldsymbol{\phi}$: $g_{\boldsymbol{\phi}}$ maps perturbation descriptors to target probabilities, and $h_{\boldsymbol{\phi}}$ maps targets and descriptors to intervention-strength parameters. In PerturbPFN, these descriptor-to-intervention maps are amortized by MLPs that take context-conditioned node and query representations as input. Given $\mathcal{I}$, the perturbation modifies only the targeted mechanisms through shift interventions (Appendix B). PerturbPFN uses this decomposition as a structural inductive bias: it predicts point estimates of $G$, $\mathbf{I}$, and $\psi$, and decodes the response through the corresponding intervened causal generative process.

## 2.2. Hierarchical Synthetic Prior

PerturbPFN is trained on prior-predictive episodes from a hierarchical synthetic SCM prior tailored to single-cell perturbation data. At the episode level, the prior samples a gene-regulatory network (GRN) using a generator inspired by Aguirre et al. (2025), augmented with module structure (Bartlett et al., 2026). At the perturbation level, descriptors $\gamma$ generate sparse atomic targets from a truncated geometric distribution target-count prior, with similar descriptors tending to produce related targets and effect strengths. Intervention strengths are dose-dependent through Hill functions, and their effects are propagated through the latent graph. Observations are then sampled from a zero-inflated log-normal expression model with SERGIO noise injection (Dibaeinia & Sinha, 2020). We reuse exogenous noise across factual and intervened synthetic rollouts to reduce sampling variance in counterfactual supervision (Sextro et al., 2026). See Appendix C for further details.

## 2.3. PerturbPFN Architecture

PerturbPFN is a transformer-based amortized predictor implementing the latent factorization above. Given context observations and a query regime, it first constructs episode-level representations and predicts an episode-level graph. It then infers targets and strengths for the context regimes, encodes these inferred interventions as regime-level memory tokens, and lets the query representation attend to this memory before predicting query-specific intervention variables. Finally, the predicted graph and query intervention are passed to an intervened SCM decoder to obtain a response prediction.

**Episode representation.** A shared context encoder processes perturbation–response pairs $(\gamma_i, \mathbf{x}_i)$ and produces node-level and global episode summaries for target, strength, and outcome prediction. In parallel, the structure branch uses a set-to-graph encoder (Dhir et al., 2025) to infer graph-specific representations that are permutation-invariant over context observations:

$$H^G = E_\eta^G(\mathcal{D}_{\text{ctx}}, \gamma_j) = \{h_1^G, \dots, h_p^G, h_{\text{glob}}^G\}.$$

**Structure head and DAG projection.** For each ordered pair $u \neq v$, the structure head predicts

$$\ell_{uv} = f_{\text{edge}}(h_u^G, h_v^G, h_{\text{glob}}^G), \quad P_{uv} = \sigma(\ell_{uv}), \quad P_{uu} = 0,$$

where $P_{uv}$ is the soft weight of edge $u \to v$. We encourage acyclicity during training with a NOTEARS-style penalty (Zheng et al., 2018)

$$\mathcal{L}_{\text{dag}} = \text{tr}(\exp(P \odot P/p)) - p.$$

For hard graph extraction, we apply $\hat{G} = \Pi_{\text{DAG}}(P; \tau)$, which thresholds candidate edges and greedily adds them

in decreasing score order while skipping edges that create cycles. Thus the graph is guaranteed to be acyclic.

**Intervention heads and SCM rollout.** For each query $\gamma_j$, intervention heads predict target probabilities $\hat{\mathbf{I}}_j$ and strengths $\hat{\boldsymbol{\psi}}_j$. The SCM decoder rolls out variables in the topological order of the projected graph. Let $r_u$ be the generated representation of variable $u$ and $m_v$ the parent message for variable $v$. Hard and soft rollouts use

$$m_v^{\text{hard}} = \sum_{u \in \text{Pa}_{\hat{G}}(v)} r_u, \quad m_v^{\text{soft}} = \sum_{u \in \text{Pa}_{\hat{G}}(v)} P_{uv} r_u.$$

The soft rollout is used during the final training stage, while test-time inference uses the hard projected graph. Each local nonlinear mechanism receives $m_v$, target indicators, and strengths, and outputs $\hat{p}_j(\mathbf{x})$. This forces high-dimensional responses through sparse local interventions and graph propagation rather than an unstructured regression head.

## 2.4. Training Curriculum and Inference

As shown in Figure 1, PerturbPFN uses synthetic labels available by construction to train a multi-task objective over graph recovery, target identification, strength prediction, acyclicity regularization, and outcome likelihood:

$$\mathcal{L} = \lambda_G \mathcal{L}_G + \lambda_I \mathcal{L}_I + \lambda_\psi \mathcal{L}_\psi + \mathcal{L}_{\text{obs}} + \lambda_{\text{dag}} \mathcal{L}_{\text{dag}}.$$

Here $\mathcal{L}_G$ is an edge-level graph loss, $\mathcal{L}_I$ is target binary cross-entropy, $\mathcal{L}_\psi$ supervises intervention strengths using a target-aware bar-distribution loss, and $\mathcal{L}_{\text{obs}}$ is the negative log-likelihood of the observed response under the decoder distribution. Training uses a staged curriculum to stabilize the interaction between graph discovery and effect prediction. We first pretrain the structure branch, then train the full model while rolling out the decoder on the ground-truth graph, and finally switch to the predicted soft graph defined above. At test time, PerturbPFN performs a single forward pass on the context episode and query descriptor, extracts a hard graph by thresholding and DAG projection, and rolls out the decoder using this hard point-estimate graph; no parameters are updated on the new episode. Architecture and training details are in Appendix D.

## 3. Experiments

We evaluate PerturbPFN along three axes: in-prior latent recovery, real perturbation effect prediction, and GRN structure discovery. Together, these tests assess whether a single synthetic-only checkpoint learns a non-collapsed graph–target–strength bottleneck, predicts held-out perturbation responses, and transfers useful regulatory signal to external biological data. All evaluations use **the same pretrained checkpoint** without task-specific gradient updates.

*Table 1.* In-prior validation over 128 held-out synthetic batches.

| Graph | | Target | | Strength / Effect | |
|---|---|---|---|---|---|
| AUROC | 0.8133 | AUROC | 0.9330 | $\psi$ ratio | 0.9386 |
| AUPRC | 0.2980 | AUPRC | 0.7860 | Delta cos. | 0.7437 |
| Density | 0.0363 | F1 | 0.7649 | Obs. NLL | -0.1375 |

*Table 2.* Average runtime per held-out protocol, normalized effect-prediction scores, and overall ranking across metrics.

| Model | Time (s) / protocol | $\hat{s}_{MD}$ ↑ | $\hat{s}_r$ ↑ | $\hat{s}_{W2}$ ↑ | Avg. $\hat{s}$ / rank |
|---|---|---|---|---|---|
| RF X-Learner | $7.20 \pm 1.64$ | 0.8793 | 0.9696 | 0.8122 | **0.8870** (#1) |
| CondOT | $6826.5 \pm 37.4$ | 0.8014 | 0.9323 | 0.7401 | **0.8246** (#3) |
| MLP | $3.19 \pm 1.08$ | 0.7823 | 0.9216 | 0.7582 | **0.8207** (#4) |
| GIM | $3994.4 \pm 587.3$ | 0.6909 | 0.8611 | 0.2934 | **0.6151** (#5) |
| Naive | / | 0.5435 | 0.6293 | 0.6299 | **0.6009** (#6) |
| Observational | / | 0.0000 | 0.0000 | 0.0698 | **0.0233** (#7) |
| PerturbPFN (amortized) | $833.9 \pm 1.87$ | 0.7186 | 0.8813 | 0.9183 | **0.8394** (#2) |
| PerturbPFN (ICL only) | $8.21 \pm 1.87$ | | | | |

### 3.1. In-Prior Latent Recovery

We first check whether the learned latent bottleneck is non-collapsed under the synthetic prior in Table 1. Since synthetic episodes provide ground-truth graphs, targets, strengths, and outcomes by construction, this evaluation measures whether PerturbPFN recovers the latent variables it is trained to infer before transferring to real data.

### 3.2. Perturbation Effect Prediction

We next evaluate whether PerturbPFN can predict held-out perturbation responses in real single-cell data. We use the Sci-Plex drug perturbation dataset of Srivatsan et al. (2020) under a highest-dosage holdout protocol following Schneider et al. (2025): for each held-out cell line–drug pair, the model observes lower doses of that drug and all doses of the remaining drugs, and must predict the response distribution at the held-out highest dose. We evaluate three complementary metrics: mean distance (MD), Pearson correlation $r$, and Wasserstein distance $W_2 = W(\hat{X}_{int}, X_{int})$. For each protocol and metric, we min–max normalize model performance after orienting the metric so that scores lie in $[0, 1]$, and then average the three normalized scores to obtain the overall rank in Table 2. Appendix Figure 2 shows the raw metric distributions across the 12 held-out protocols.

Table 2 shows that PerturbPFN achieves the best normalized $W_2$ score and ranks second on the three-metric average, behind RF X-Learner. This indicates that the model provides competitive distributional prediction while retaining explicit graph, target, and strength estimates. However, PerturbPFN does not uniformly dominate real-data supervised baselines: RF X-Learner and MLP remain stronger on mean-response and directional metrics. Thus, the main advantage of PerturbPFN is a complementary trade-off between structured latent inference, low test-time adaptation cost, and competitive held-out response prediction.

*Table 3.* GeneRNIB 300BCG GRN recovery. Higher is better for all metrics.

| Model | GS-F1 ↑ | TFB-F1 ↑ | R-Prec. ↑ | R-Recall ↑ | VC ↑ |
|---|---|---|---|---|---|
| Pearson corr. | 0.4693 | 0.1102 | **0.6154** | 0.2742 | **0.1882** |
| PPCOR | 0.0982 | 0.0000 | 0.1742 | 0.1920 | 0.0660 |
| GRNBoost | 0.5566 | 0.1129 | 0.6082 | **0.3752** | 0.1576 |
| Portia | 0.6188 | 0.1014 | 0.3970 | 0.2668 | 0.1169 |
| SCENIC | 0.4813 | **0.1180** | 0.3802 | 0.2747 | 0.0672 |
| PerturbPFN | **0.6410** | 0.1146 | 0.2250 | 0.1977 | 0.0731 |

### 3.3. GRN Structure Discovery

We further evaluate whether the learned graph head transfers to external biological GRN recovery. We use the official GeneRNIB 300BCG benchmark and compare against standard GRN baselines (Nourisa et al., 2025) in Table 3. PerturbPFN achieves the best GS-F1 and competitive TFB-F1, suggesting transferable regulatory signal, but it underperforms dedicated GRN baselines on R-Precision, R-Recall, and VC. This is expected because PerturbPFN is trained to infer an intervention-induced causal graph, which is related to but not identical to a curated gene-regulatory network. A GRN benchmark may emphasize TF–target annotations, gene-set overlap, or other biological reference signals, whereas PerturbPFN's graph is optimized to support perturbation propagation under a synthetic SCM prior.

Compared with conventional causal discovery methods such as BaCaDi* (Hägele et al., 2023) and JCI-PC (Mooij et al., 2020; Spirtes et al., 2000), PerturbPFN can amortize structural inference from synthetic biological priors. GRNs exhibit sparsity, modular organization, degree heterogeneity, and richer topological regularities captured by specialized GRN generators (Aguirre et al., 2025; Dibaeinia & Sinha, 2020; Barabási & Albert, 1999). By training on samples from such generators, PerturbPFN can exploit simulator-defined biological priors, which is a key strength of PFNs (Müller et al., 2025). Consistent with this interpretation, PerturbPFN performs strongly on direct graph-ranking diagnostics, achieving edge AUROC 0.9168 and edge average precision 0.7570, although these edge-ranking metrics are not part of the official GeneRNIB score table.

## 4. Discussion

PerturbPFN demonstrates that PFN-style training on structured synthetic biological priors can amortize unknown-target perturbation prediction while exposing interpretable estimates of targets, strengths, and system structure. Although it does not uniformly dominate specialized baselines, it offers a useful trade-off: competitive prediction with low in-context inference cost and no dataset-specific optimization. Future work should improve uncertainty estimation, simulator fidelity, and scaling to larger gene panels.

## Acknowledgements

We thank the workshop organizers and anonymous reviewers for their insightful comments and suggestions. We gratefully acknowledge GPU resources provided by the Computational and Biological Learning Lab (CBL), University of Cambridge. Y.G. gratefully acknowledges support from the Student Educational Development Award from St Edmund's College, University of Cambridge.

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

# A. Related Work

**Perturbation response prediction.** Many single-cell perturbation models treat the task as supervised counterfactual response prediction. Autoencoder and latent-variable methods such as scGen (Lotfollahi et al., 2019), CPA (Lotfollahi et al., 2023), Biolord (Piran et al., 2024), and SAMS-VAE (Bereket & Karaletsos, 2023) learn perturbation or attribute representations that are recombined with cell-state embeddings to decode post-perturbation expression profiles, with ChemCPA additionally using chemical descriptors for unseen-drug prediction (Hetzel et al., 2022). Optimal-transport methods such as CellOT (Bunne et al., 2023) and CondOT (Bunne et al., 2022) instead learn maps between unpaired control and treated-cell distributions, optionally conditioned on perturbation context. Knowledge-informed predictors such as GEARS incorporate gene–gene graphs with graph neural networks for genetic perturbation prediction (Roohani et al., 2024). These methods can be effective, but they typically predict responses end-to-end through latent spaces, transport maps, or external knowledge graphs, rather than jointly inferring unknown atomic targets, intervention strengths, and propagation through an inferred SCM.

**Structured perturbation models.** A complementary line of work represents perturbations as interventions on structured or causal models, aiming to recover mechanisms rather than only predict expression shifts. Classical multi-environment causal discovery methods, including JCI-PC (Mooij et al., 2020), UT-IGSP (Squires et al., 2020), and BaCaDi (Hägele et al., 2023), infer causal graphs and, in some cases, unknown intervention targets from observational and interventional data. More recent perturbation models make this link explicit: GIM learns a generative map from perturbation descriptors to atomic interventions in a causal model (Schneider et al., 2025), while SCCVAE combines a variational autoencoder with a learned regulatory network and shift interventions for genetic perturbation response prediction (Liu et al., 2026). Related inverse-design methods such as PDGrapher use causally inspired graph neural networks to predict therapeutic target sets that would move a diseased state toward a desired treated state (Gonzalez et al., 2025). These approaches provide stronger mechanistic structure than end-to-end predictors, but their reliance on instance-specific fitting, iterative inference, or predefined proxy graphs increases computational cost and limits amortized transfer across new perturbation contexts.

**Prior-data fitted networks and single-cell foundation models.** Prior-data fitted networks (PFNs) show that transformers pretrained on synthetic tasks can amortize Bayesian-style inference and make predictions on new datasets by conditioning on examples in context, motivating recent tabular foundation models such as TabPFN and TabICL (Müller et al., 2022; Hollmann et al., 2025; Qu et al., 2026). This paradigm has recently been extended to causal inference, including amortized causal effect estimation (Robertson et al., 2025; Balazadeh et al., 2025), interventional prediction under graph uncertainty (Dhir et al., 2026), and general PFN frameworks for causal reasoning (Ma et al., 2025); MapPFN is especially related, learning causal perturbation maps for biological response prediction in context (Sextro et al., 2026). In parallel, single-cell foundation models such as scGPT (Cui et al., 2023), Geneformer (Theodoris et al., 2023), and GenePT (Chen & Zou, 2023) learn transferable gene or cell representations from transcriptomic corpora or biomedical text, while perturbation-oriented models such as LPM (Miladinovic et al., 2025) and Stack (Dong et al., 2026) use symbolic perturbation–readout–context tuples or in-context sets of cells for response prediction. PerturbPFN combines these directions: it uses PFN-style synthetic-prior pretraining, but specializes the prior to structured perturbation mechanisms, explicitly predicting latent graphs, unknown targets, and intervention strengths before decoding the perturbation response.

# B. SCM Background and Shift Interventions

A structural causal model (SCM) $\mathcal{M} = \{G, \boldsymbol{\theta}\}$ represents a system by a directed acyclic graph (DAG) with adjacency matrix $G \in \{0,1\}^{p \times p}$ over variables $\mathbf{x} = (x_1, \ldots, x_p)$ and a collection of local mechanisms parameterized by $\boldsymbol{\theta}$. The induced observational distribution factorizes as

$$p(\mathbf{x}; G, \boldsymbol{\theta}) = \prod_{i=1}^{p} p_i(x_i \mid \mathbf{x}_{G_i}; \boldsymbol{\theta}_i),$$

where $G_i$ denotes the parent set of variable $i$. This factorization is useful because modifying one local mechanism can change the distribution of its target variable and propagate to downstream descendants through the graph.

An intervention on an SCM modifies a subset of local mechanisms. A general shift intervention changes the targeted mechanisms by applying a local shift or modulation,

$$p_i(x_i \mid \mathbf{x}_{G_i}; \boldsymbol{\theta}_i) \quad \longrightarrow \quad \tilde{p}_i(x_i \mid \mathbf{x}_{G_i}; \boldsymbol{\theta}_i, \psi_i), \qquad i \in I,$$

while leaving non-targeted mechanisms unchanged. Unlike hard interventions, which set variables externally and remove their dependence on parents, shift interventions retain parent-dependent regulation. In this work, we use additive mechanism shifts in the latent SCM. Writing a local mechanism as

$$x_i = f_i(\mathbf{x}_{G_i}; \boldsymbol{\theta}_i) + \epsilon_i,$$

a targeted intervention with strength $\psi_i$ changes it to

$$x_i = f_i(\mathbf{x}_{G_i}; \boldsymbol{\theta}_i) + \psi_i + \epsilon_i, \qquad i \in I,$$

with $\psi_i = 0$ for non-targeted variables. The post-intervention distribution is obtained by rolling out the modified SCM in topological order.

## C. Details on Synthetic Data Generation

This section provides additional details on the hierarchical synthetic SCM prior used to pretrain PerturbPFN. Each synthetic episode is sampled in three levels: an episode-level gene-regulatory system, perturbation-level atomic interventions, and observation-level single-cell responses.

*Table 4.* Main hyperparameters for the module-structured GRN topology prior used in synthetic data generation.

| Symbol | Description | Range / value |
|---|---|---|
| $p$ | Number of genes / nodes | $[40, 60]$ |
| $K$ | Number of gene modules | $[4, 8]$ |
| $\alpha_0$ | Base edge logit | $-2.5$ |
| $\sigma_s$ | Source activity log-normal scale | $0.9$ |
| $\alpha_t$ | Target susceptibility Gamma shape | $2.0$ |
| $\mu_B, \sigma_B$ | Module-pair logit mean and scale | $-0.2, 0.35$ |
| $\beta_{\text{within}}$ | Within-module logit boost | $1.35$ |
| $\lambda_{\text{dist}}$ | Topological distance penalty | $1.5$ |

**Episode-level GRN.** Our GRN prior is inspired by the structural properties emphasized by Aguirre et al. (2025): real gene-regulatory networks are sparse, directed, degree-heterogeneous, and modular. However, because our SCM decoder requires ancestral rollout, we instantiate these properties through an acyclic module-structured DAG prior rather than directly using a cyclic preferential-attachment generator. Table 4 summarizes the main topology hyperparameters. For each episode, we first sample the number of genes $p$ and assign genes to $K$ balanced modules. We then impose a topological order and only allow edges from earlier to later nodes. For a candidate edge $u \to v$, $u < v$, we sample

$$A_{uv} \sim \text{Bernoulli}(\sigma(\ell_{uv})),$$

with

$$\ell_{uv} = \alpha_0 + \log s_u + 0.35 \log t_v + B_{c_u, c_v} - \lambda_{\text{dist}} \frac{v - u}{p - 1},$$

where $s_u$ is a source-specific regulatory activity, $t_v$ is a target-specific susceptibility, $c_u, c_v$ are module assignments, and $B$ is a module-pair affinity matrix with an additional within-module boost. This construction induces sparse, directed, degree-heterogeneous, and modular GRN-like DAGs while guaranteeing acyclicity.

**Perturbation descriptors and targets.** At the perturbation level, each episode contains a set of synthetic drugs organized into latent drug families. We use a bipartite family–module affinity prior: each drug family is assigned high affinity to a small number of preferred gene modules and low affinity to the remaining modules. Drugs inherit a family embedding with drug-specific noise, so nearby descriptors tend to correspond to related target modules. For drug $d$, the target propensity of gene $i$ is proportional to

$$\pi_{d,i} \propto A_{f_d, c_i} \, q_i \, \xi_{d,i},$$

where $f_d$ is the drug family, $c_i$ is the gene module, $A_{f_d, c_i}$ is the family–module affinity, $q_i$ is a gene-specific targetability factor, and $\xi_{d,i}$ is drug-specific multiplicative noise. The target set size is sampled from a truncated geometric distribution, and targets are sampled without replacement according to $\pi_{d,i}$. This construction creates sparse interventions while making related drug descriptors more likely to share targets and effect patterns.

*Table 5.* Main hyperparameters for perturbation, dose-response, and observation sampling.

| Symbol | Description | Range / value |
|--------|-------------|---------------|
| $F$ | Number of drug families | $[3, 6]$ |
| $D$ | Number of synthetic drugs | $[12, 24]$ |
| $d_{\text{emb}}$ | Drug embedding dimension | 8 |
| $M_{\text{max}}$ | Max preferred modules per family | 2 |
| $p_{\text{stop}}$ | Geometric target-count stop probability | 0.08 |
| $\lambda_{\text{target}}$ | Gene targetability strength | 0.15 |
| $\sigma_{\text{drug}}$ | Drug-specific log-propensity noise | 0.55 |
| $a_{\text{min}}, a_{\text{max}}$ | Dose range | $[0.02, 10.0]$ |
| $n_{\text{min}}, n_{\text{max}}$ | Hill coefficient range | $[2.0, 5.0]$ |
| $\sigma_{\text{obs}}$ | Log-normal observation noise range | $[0.05, 0.15]$ |
| $C$ | Simulated cells per condition | 128 |

**Dose-dependent intervention strengths.** For a drug $d$, dose $a$, and target gene $i$, the intervention strength is generated by a signed Hill response,

$$\psi_{d,a,i} = s_{d,i} \, E_{d,i}^{\text{max}} \frac{a^{n_{d,i}}}{(EC50_{d,i})^{n_{d,i}} + a^{n_{d,i}}},$$

where $s_{d,i} \in \{-1, 1\}$ is the effect sign, $E_{d,i}^{\text{max}}$ is the maximum effect size, $EC50_{d,i}$ is the half-maximal dose, and $n_{d,i}$ is the Hill coefficient. Doses are sampled log-uniformly from $[a_{\text{min}}, a_{\text{max}}]$. Non-targeted genes have $\psi_{d,a,i} = 0$, and control conditions have zero target masks and zero strengths.

**SCM rollout and additive shifts.** Given a sampled DAG and intervention, latent expression is generated in topological order. For each gene, the unperturbed local mechanism combines a gene-specific basal expression parameter with nonlinear Hill-type responses from its parents. The intervention applies an additive shift to targeted local mechanisms:

$$z_i = f_i(\mathbf{z}_{G_i}; \boldsymbol{\theta}_i) + I_i \psi_i + \epsilon_i,$$

where $z_i$ denotes latent expression before observation noise, $I_i$ is the target indicator, and $\epsilon_i$ is exogenous noise. Because descendants are generated after their parents, shifts to targeted genes propagate downstream through the sampled GRN.

**Observation model.** To mimic single-cell expression measurements, latent expression is transformed into observed expression using a zero-inflated log-normal model (Dibaeinia & Sinha, 2020). Positive observations are sampled by adding Gaussian noise in log space, with gene-specific log-normal noise scales. Dropout is sampled from an expression-dependent Bernoulli variable, so genes with lower latent expression are more likely to be observed as zeros. This produces sparse, heteroscedastic expression vectors resembling single-cell readouts.

**Episode construction and paired rollouts.** Each training episode contains a context set and a query set sampled from the same latent task. The context includes a control condition and additional perturbation conditions, while query conditions are held out from the context. Within an episode, conditions reuse the same exogenous base-noise matrix for latent SCM rollout. In the SCM sense, this creates paired counterfactual rollouts: different interventions are applied to the same underlying latent cell noise. This reduces variance in intervention-induced differences and provides cleaner supervision for targets, strengths, and effect prediction.

## D. Model Architecture and Training

This section summarizes the implementation details. Table 6 gives the main architecture dimensions, and Table 7 gives the key training settings.

The shared encoder processes context perturbation–response pairs and produces node-level and global episode summaries for target, strength, and outcome prediction. The structure encoder separately produces graph-specific node representations for edge scoring. For each ordered pair $u \neq v$, the structure head combines a bilinear source–destination score with an MLP over pairwise node features:

$$\ell_{uv} = \frac{(W_s h_u^G)^\top (W_t h_v^G)}{\sqrt{d_h}} + \text{MLP}_{\text{edge}}\left([W_s h_u^G, \, W_t h_v^G, \, (W_s h_u^G) \odot (W_t h_v^G), \, W_s h_u^G - W_t h_v^G]\right),$$

*Table 6.* Main architecture settings for PerturbPFN.

| Component | Setting | Value |
|---|---|---|
| Input graph size | Maximum nodes | 60 |
| Regime descriptor | Metadata dimension | 11 |
| Shared encoder | Hidden dimension | 192 |
| Shared encoder | Attention heads / blocks | 6/5 |
| Shared encoder | Feedforward dimension | 640 |
| Structure encoder | Hidden dimension | 256 |
| Structure encoder | Attention heads / blocks | 8/4 |
| Heads | MLP hidden dimension | 320 |
| SCM decoder | Hidden dimension / layers | 64/2 |
| SCM decoder | Activation | tanh |

followed by $P_{uv} = \sigma(\ell_{uv})$ and $P_{uu} = 0$. Hard graph extraction uses the DAG projection in Section 2.3: candidate edges with $P_{uv} \geq \tau$ are sorted in decreasing $P_{uv}$, and each edge is added only if it does not create a directed cycle.

The target head predicts node-wise target probabilities; conditioned on these target scores, the strength head predicts node-wise intervention strengths. The SCM decoder rolls out variables in topological order and outputs zero-inflated log-normal parameters for each query response. In hard rollout, parent messages are aggregated over the binary parent set of the projected DAG. In soft rollout, the projected DAG still determines the topological order and admissible parent set, but parent contributions are weighted by the continuous edge probabilities $P_{uv}$.

*Table 7.* Main training settings.

| Category | Setting | Value |
|---|---|---|
| Synthetic batch | Batch size | 8 |
| Synthetic batch | Context / query regimes | 10/4 |
| Optimizer | Optimizer | AdamW |
| Optimizer | Learning rate | $3 \times 10^{-4}$ |
| Optimizer | Weight decay | $10^{-4}$ |
| Optimizer | Gradient clipping | 1.0 |
| Training length | Total synthetic steps | 7000 |

Training follows the staged curriculum in Section 2.4. During structure pretraining, only the structure encoder and structure head are optimized. During graph-GT joint training, the full model is optimized while the decoder rolls out on the ground-truth synthetic graph. During soft-graph joint training, the decoder uses the projected predicted graph topology with continuous edge probabilities as parent weights, exposing the outcome model to its own graph predictions before test time. At inference time, PerturbPFN performs a single forward pass, thresholds edge probabilities at $\tau = 0.5$, applies the greedy DAG projection above, and rolls out the decoder using the hard projected graph.

## E. Details on Effect Prediction

This section provides additional details on the real-data effect-prediction benchmark, including the dataset preprocessing, highest-dosage holdout protocol, evaluation metrics, and baseline implementations.

### E.1. Dataset

We evaluate on the Sci-Plex single-cell drug perturbation dataset of Srivatsan et al. (2020), following the preprocessed epigenetic-regulation benchmark used by Schneider et al. (2025). The benchmark contains three cell lines, A549, K562, and MCF7, and four small-molecule perturbations, belinostat, dacinostat, givinostat, and quisinostat 2HCl, giving $3 \times 4 = 12$ held-out protocols. Each protocol is defined by one cell line–drug pair.

We use the same preprocessing convention as the GIM benchmark. Cells are normalized to the median count of control cells, filtered, log-transformed, and restricted to a protocol-specific set of 50 marker genes selected using only the training conditions. Perturbation descriptors encode drug identity and dose information. Control cells are included as the observational regime.

### E.2. Highest-Dosage Holdout Protocol

For each cell line–drug protocol, the highest dose of the held-out drug is removed from the context and used as the query condition. The context set contains control cells, lower doses of the held-out drug, and all available doses of the remaining drugs in the same cell line. The model must predict the distribution of expression vectors under the held-out highest dose.

This protocol tests dose extrapolation under unknown perturbation targets. The held-out drug is not completely unseen, since lower-dose responses are available, but the target and mechanism at the highest dose are not provided to the model. All methods are evaluated on the same held-out cells and with the same metric implementation.

### E.3. Metrics

Let $X_{\text{ctrl}}$, $X_{\text{int}}$, and $\hat{X}_{\text{int}}$ denote control cells, true held-out perturbed cells, and predicted perturbed cells, respectively. We evaluate four complementary quantities.

**Mean distance.**    We report mean-response accuracy using mean distance,

$$\text{MD} = \left\| \mathbb{E}[\hat{X}_{\text{int}}] - \mathbb{E}[X_{\text{int}}] \right\|_2,$$

with lower values indicating more accurate prediction of the average response. The raw evaluation files also store population RMSE, computed by comparing the predicted mean response against held-out cells, but the main results use MD because it directly measures mean-response error without mixing in within-condition cellular variability.

**Pearson correlation.**    We compute Pearson correlation between the predicted and true mean response vectors,

$$r = \text{corr}\left( \mathbb{E}[\hat{X}_{\text{int}}], \mathbb{E}[X_{\text{int}}] \right),$$

which measures whether the predicted perturbation direction is correct.

**Wasserstein distance.**    We compute a Sinkhorn divergence between the predicted and true held-out cell distributions,

$$W(\hat{X}_{\text{int}}, X_{\text{int}}),$$

using entropic regularization $\epsilon = 0.1$, 200 Sinkhorn iterations, and tolerance $10^{-6}$. This metric evaluates distribution-level agreement beyond the mean response.

**Magnitude ratio.**    Following Sextro et al. (2026), we compute the Wasserstein magnitude ratio

$$\text{MagRatio} = \frac{W(X_{\text{ctrl}}, \hat{X}_{\text{int}})}{W(X_{\text{ctrl}}, X_{\text{int}})}.$$

A value near 1 indicates that the predicted perturbation has the correct effect magnitude; values below 1 indicate under-response toward the control distribution, and values above 1 indicate over-amplification.

**Normalized multi-metric score.**    For the normalized multi-metric summary in Table 2, each metric is oriented so that higher is better and min–max normalized within each protocol:

$$s_{m,t,k} = \frac{z_{m,t,k} - z_{t,k}^{\min}}{z_{t,k}^{\max} - z_{t,k}^{\min}},$$

where $m$ indexes models, $t$ indexes held-out protocols, $k$ indexes metrics, and $z$ denotes the oriented metric value. We average the four normalized scores to obtain the final score and rank.

### E.4. Baselines

We compare against structured causal, distributional, supervised, and heuristic baselines. All baselines are evaluated on the same held-out protocols, and all reported metrics are recomputed using the same metric code as PerturbPFN whenever prediction arrays are available.

**GIM.** GIM (Schneider et al., 2025) is the closest structured causal baseline. We use the authors' implementation and Sci-Plex preprocessing. GIM is fit separately for each held-out protocol by instance-specific optimization. We use the Sci-Plex configuration from the GIM codebase: a nonlinear zero-inflated log-normal inference model, unknown intervention targets, Adam optimization with step size $10^{-3}$, graph prior and intervention sparsity penalties, and augmented-Lagrangian acyclicity optimization. The main GIM runs use 100,000 optimization steps following the original setting.

**CondOT.** CondOT (Bunne et al., 2022) is an optimal-transport baseline that learns a conditional transport map from control cells to perturbed cells. We use the CondOT/PICNN-style implementation and settings used in the GIM benchmark, with input labels given by the perturbation descriptor. The transport networks use four hidden layers with 128 hidden units. For the final reported runs, CondOT is fit separately for each protocol using the same training conditions as the other real-data baselines, and predictions are generated by transporting sampled control cells to the held-out highest-dose condition.

**MLP.** The MLP baseline follows the shift-regression baseline from the GIM codebase. It learns a mapping from perturbation descriptors to mean expression shifts relative to the control distribution. At prediction time, the estimated shift for the held-out descriptor is added to sampled control cells to form predicted perturbed cells. We use a two-layer MLP regressor with hidden size 100, learning rate $10^{-3}$, and maximum 20,000 iterations.

**RF X-Learner.** RF X-Learner is a supervised treatment-effect baseline based on the X-learner meta-learning strategy (Künzel et al., 2019). The model first fits a global response model $\mu(\gamma)$, then constructs imputed treatment effects for treated and control samples and fits two effect models $\tau_1(\gamma)$ and $\tau_0(\gamma)$. At prediction time, the final effect is a global constant-weight mixture of $\tau_0$ and $\tau_1$, added to sampled control cells to generate the predicted perturbed distribution. The backbone is a multi-output `RandomForestRegressor` with 100 trees, no maximum depth, full training data, and no downsampling; targets are z-scored during fitting and transformed back to the original expression scale for evaluation.

**Naive and Observational.** The Naive baseline predicts the held-out highest-dose response using the nearest observed lower dose of the same drug in the same cell line. The Observational baseline ignores the perturbation and predicts the control distribution. These baselines require no training and serve as heuristic lower bounds for dose extrapolation and perturbation response modelling.

**Boxplot Results.** See Figure 2 for distribution of raw metric values across the 12 held-out protocols.

## F. Details on Structure Discovery

This section provides additional details on the synthetic structure-discovery protocol, graph metrics, baseline implementations, and the graph-size generalization analysis.

### F.1. Protocol

We evaluate structure discovery on a fixed in-prior synthetic benchmark designed to test graph recovery under the same module-structured SCM family used to pretrain PerturbPFN. Each dataset contains $p = 60$ measured variables, 16 synthetic drugs, one control regime, and one treated regime per drug. Each regime contains 128 simulated cells. We generate 20 independent synthetic datasets using different random seeds.

The ground-truth graph is a module-structured DAG sampled from the PerturbPFN synthetic prior. Perturbations are soft shift interventions with sparse unknown targets. The target count is sampled from a shifted geometric prior with stop probability $1/3$, giving an expected target count of approximately 3. The mechanism family is nonlinear, and observations are sampled from the zero-inflated log-normal expression model used in pretraining.

For PerturbPFN, we evaluate the same final checkpoint used in the main experiments. For each synthetic dataset, we sample 64 context/query episodes, average the predicted edge probabilities across episodes, threshold edge probabilities at $\tau = 0.5$, and apply the deterministic DAG projection described in Section 2.3. The projected hard DAG is then compared to the ground-truth graph. The threshold is fixed by the protocol and is not tuned per dataset.

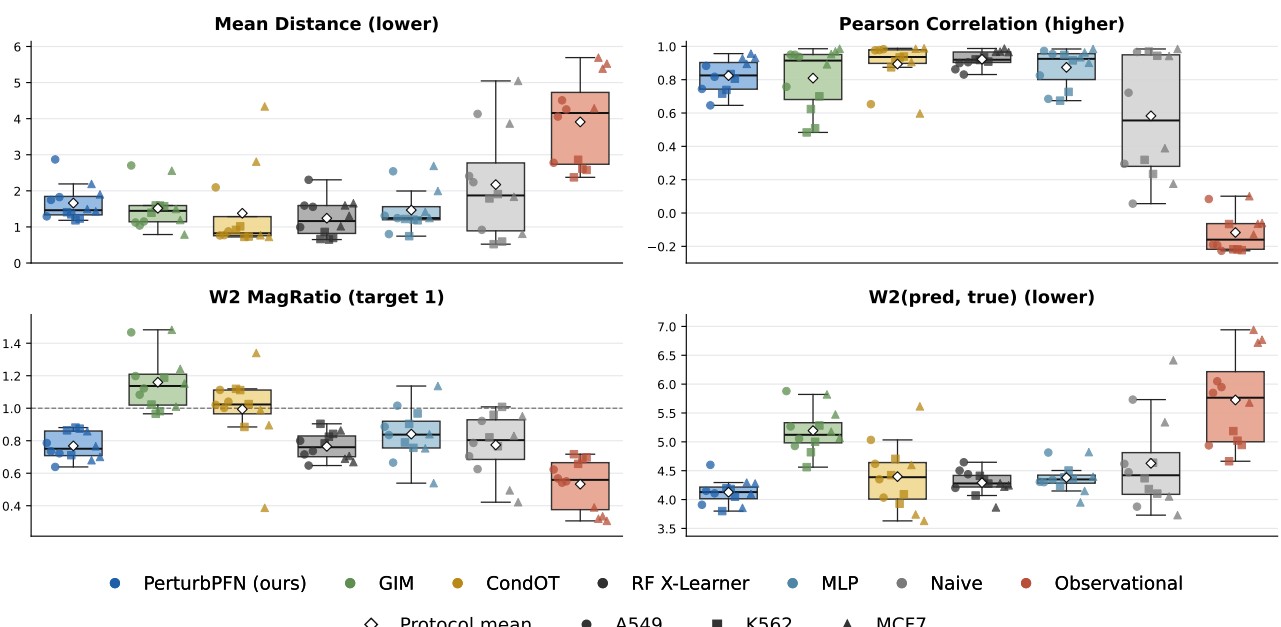

*Figure 2.* Effect prediction on the 12-protocol highest-dosage holdout benchmark from Srivatsan et al. (2020). Each point corresponds to one held-out cell line–drug protocol. PerturbPFN achieves competitive mean-distance and effect-magnitude recovery, with stable Pearson correlation across protocols, while avoiding the magnitude underestimation observed for RF X-Learner and the over-amplification observed for GIM.

### F.2. Metrics

We report the same graph-recovery metrics for PerturbPFN and all baselines. All metrics are computed on directed adjacency matrices after applying each method's graph extraction rule.

**Structural Hamming distance.** Structural Hamming distance (SHD) counts edge additions, deletions, and orientation errors between the predicted graph and the ground-truth DAG. Lower SHD indicates closer graph recovery.

**Structural intervention distance.** Structural intervention distance (SID) compares the predicted graph $\hat{G}$ to the ground-truth graph $G$ by counting ordered node pairs for which $\hat{G}$ gives an incorrect adjustment set for estimating the causal effect in $G$ (Peters & Bühlmann, 2015; Peters et al., 2017). Informally,

$$\text{SID}(\hat{G}, G) = \#\{(i, j) : i \neq j, \ \hat{G} \text{ implies an invalid adjustment for } i \rightarrow j \text{ in } G\}.$$

Lower SID indicates that the predicted graph better preserves the interventional implications of the ground-truth DAG. We compute SID using the CDT implementation.

**Directed edge recovery.** We compute edge precision, recall, and F1 on the directed adjacency matrices. Precision measures the fraction of predicted edges that are correct, recall measures the fraction of ground-truth edges recovered, and F1 is their harmonic mean.

**Acyclicity.** The DAG metric is an indicator for whether the predicted graph is acyclic. For methods that output partially directed graphs, we use the same baseline evaluation code to orient undirected edges before computing directed metrics. All reported values are averaged over the 20 synthetic datasets.

### F.3. Baselines

We compare against causal discovery baselines implemented in the GIM codebase, using the same input datapackages and graph-metric evaluation pipeline for all methods. Each baseline is fit separately on each synthetic dataset.

**BaCaDi\*.** BaCaDi* (Hägele et al., 2023) is run as an instance-specific Bayesian causal discovery baseline with unknown intervention targets and shift-intervention support. We use a nonlinear inference model with Adam optimization, augmented-Lagrangian acyclicity optimization, an Erdős–Rényi graph prior with two expected edges per node, and 30,000 optimization steps. We tune the graph and intervention sparsity penalties for this protocol and use the resulting hard graph for evaluation.

**JCI-PC and JCI-PC with context variables.** JCI-PC (Mooij et al., 2020) is a constraint-based joint causal inference baseline. We run two variants: one without using perturbation descriptors as context variables, and one with context variables. Both variants use Gaussian conditional independence tests. We use significance level $10^{-4}$ for JCI-PC and $10^{-3}$ for JCI-PC with context variables, selected from a small tuning sweep on this protocol.

**UT-IGSP.** UT-IGSP (Squires et al., 2020) is an unknown-target interventional structure-learning baseline. We use the Gaussian test variant with shift-intervention mode enabled. The main significance settings are $\alpha = 0.1$ and $\alpha_{\text{inv}} = 0.001$, with the same rank-check tolerance and failure handling used in the baseline suite.

**GIM.** GIM (Schneider et al., 2025) is run as an instance-specific generative intervention model on each synthetic dataset. We use a nonlinear zero-inflated log-normal inference model with unknown targets, Adam step size $10^{-3}$, augmented-Lagrangian acyclicity optimization, and sparsity regularization. For the structure discovery benchmark, GIM uses 20,000 optimization steps, an Erdős–Rényi graph prior with two expected edges per node, graph prior weight 5, and intervention sparsity weight 3.

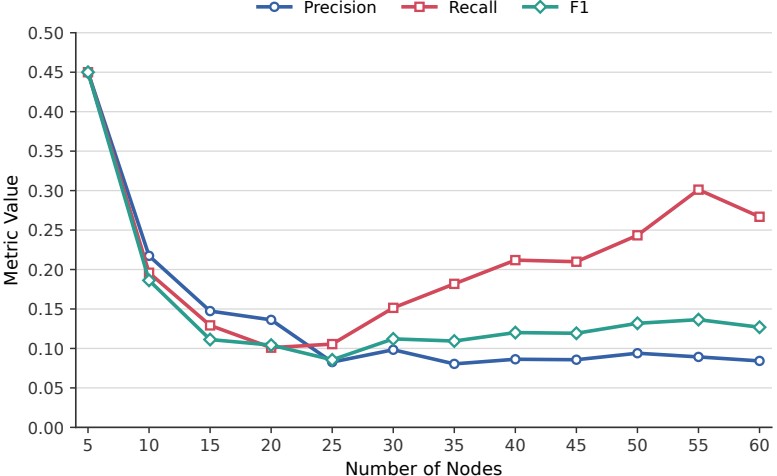

*Figure 3.* Graph-size sweep for PerturbPFN structure discovery. We evaluate the same checkpoint on synthetic structure discovery task with different numbers of nodes and report directed-edge precision, recall, and F1 after thresholding edge probabilities at 0.5 and applying DAG projection. Performance remains non-trivial across graph sizes, indicating that the learned graph-recovery behavior is not limited to the 60-node setting used in the main benchmark.

## G. Details on Target Identification

### G.1. Dataset

We construct the target-identification benchmark from the Norman19 Perturb-seq dataset (Norman et al., 2019). We restrict to single-gene perturbations, excluding combinatorial perturbations and controls from the candidate target set. A perturbation is retained only if its target gene is included in the measured gene panel and has sufficient cells for evaluation. Control cells are used as the observational regime.

### G.2. Protocol

Each benchmark task contains $p = 50$ genes and 15 single-gene perturbation regimes. The gene panel is constructed to include all true target genes and is completed with genes showing large perturbation-induced mean shifts relative to control. Perturbation labels are replaced by anonymous task-local symbols, implemented as randomly permuted one-hot regime

descriptors, so the model cannot use gene names or semantic perturbation information. For each perturbation regime, the true target identity is hidden from the model and used only for evaluation.

For PerturbPFN, the context contains sampled response cells from the anonymous regimes and controls, while query cells are used to obtain target scores for each perturbation. Target logits are averaged over query cells within each regime to produce one score vector over the 50 candidate genes. No model parameters are updated on the Norman19 task. We report means over 5 random seeds, which resample task construction and anonymous label assignments.

### G.3. Metrics

Let $s_{q,i}$ denote the target score assigned to gene $i$ for perturbation $q$, and let $t_q$ be the true knockout target.

**Top-$k$ accuracy.** Top-$k$ accuracy is the fraction of perturbations for which the true target appears among the $k$ highest-scoring genes:

$$\text{Top-}k = \frac{1}{Q} \sum_{q=1}^{Q} \mathbf{1}\{t_q \in \text{TopK}(s_q)\}.$$

**Mean reciprocal rank.** MRR evaluates the rank assigned to the true target:

$$\text{MRR} = \frac{1}{Q} \sum_{q=1}^{Q} \frac{1}{\text{rank}_q(t_q)}.$$

**F1.** For F1, target scores are converted into binary predicted target sets. For PerturbPFN, we threshold target logits at zero; for baselines with binary target outputs, we use the inferred binary target masks directly. Precision, recall, and F1 are computed over all perturbation–gene pairs.

### G.4. Baselines

All baselines are run in unknown-target mode on the same anonymous Norman19 tasks.

**GIM.** GIM (Schneider et al., 2025) fits an instance-specific generative intervention model with unknown targets. We use the nonlinear inference model from the GIM codebase and extract predicted target scores or masks for each anonymous perturbation regime.

**BaCaDi\*.** BaCaDi\* (Hägele et al., 2023) is run as an instance-specific Bayesian causal discovery and intervention-inference baseline with sparse unknown intervention targets. We use the inferred intervention-target probabilities to rank candidate genes.

**UT-IGSP.** UT-IGSP (Squires et al., 2020) is an unknown-target interventional structure-learning method. We use its inferred target sets as binary predictions and as scores for ranking.

**JCI-PC.** JCI-PC (Mooij et al., 2020; Spirtes et al., 2000) is a constraint-based joint causal inference baseline. We use the inferred intervention-target information from the learned joint graph to score candidate target genes.

