# OpenReview forum: "PerturbPFN: Probing the Limits of Synthetic Priors in Perturbation Modelling"
_ICML.cc/2026/Workshop/FMSD — FMSD @ ICML 2026 Poster_

### Official Review · Reviewer_XAm4 · 2026-05-20
**Review: Interesting idea but weak real-world results**

**Rating:** 7
**Confidence:** 2

**Review:**

### Summary

The paper proposes PerturbPFN, a response prediction model. PerturbPFN models the predictive distribution via latent intervention targets and strengths. The model is trained on synthetic episodes and evaluated on synthetic benchmarks and a real-world dataset. While the empirical results show strong structure discovery results in a synthetic benchmark, the target identification on a real-world dataset lags behind baselines.

### Strengths
- PerturbPFN models the predictive distribution via atomic targets and intervention strength rather than directly parametrizing the response of the episode.
- The paper is polished, and the methodology is clearly explained through the text and with the main figure. The appendix provides additional details, insights, and thoroughly discusses related work.

### Weaknesses
- The concept of the strengths is unclear to me. What is $\psi$ (L93 right)? A formal definition would benefit clarity.
- I suggest that the authors include pseudo-code, as the proposed methodology is composed of many different parts. Further, I recommend explaining the different heads (graph, target, strength) as they are not mentioned in the main text.
- The impact of the results is unclear. While the structure prediction performs well on synthetic data, the target identification lags behind baselines on a real-world dataset. Even though this limitation is discussed, its importance remains unclear.

---

### Official Review · Reviewer_LDEa · 2026-05-20
**A Solid Proof of Concept for PFN-Style Perturbation Modeling**

**Rating:** 6
**Confidence:** 4

**Review:**

**Summary**

This paper proposes PerturbPFN, a PFN-style perturbation model trained on biologically motivated synthetic episodes. Instead of directly regressing cellular responses, the model predicts a latent system graph, perturbation targets, and intervention strengths, and then rolls out responses through an SCM decoder. The paper evaluates this approach on real single-cell perturbation response prediction as well as synthetic benchmarks for structure discovery and target identification, and argues that the main value of the method lies in its amortized inference, low test-time cost, and interpretable intermediate predictions.

**Strengths**

1. The paper studies an interesting and well-motivated problem: predicting responses to unseen perturbations when the target mechanisms may be unknown.
2. The combination of PFN-style amortized inference with a structured SCM-based intervention bottleneck is novel and easy to understand.
3. The empirical evaluation is well organized, covering real-data effect prediction as well as synthetic evaluations for structure discovery and target identification.
4. The method exposes explicit intermediate quantities such as latent graphs, targets, and intervention strengths, rather than behaving as a pure black-box regressor.

**Areas for Improvement**

1. It would be helpful to clarify the practical regime in which PerturbPFN should be preferred over stronger specialized baselines.
2. Since interpretability is one of the paper’s main advantages, readers would benefit from a small real-data case study showing whether the inferred targets, strengths, or graph fragments align with known biology for a few representative perturbations.

**Detailed Comments**

See 1-2 in Areas for Improvement.

**Justification of Score**

The paper is well written, methodologically clear, and studies an interesting structured-data problem. I found the combination of PFN-style amortized inference and an SCM-based intervention bottleneck novel and well motivated, and the empirical evaluation is covering real-data effect prediction as well as synthetic structure and target recovery. My main reservations are that the method does not outperform the strongest specialized baselines on real response prediction, and that the practical value of the proposed trade-off and the real-data faithfulness of the intermediate mechanistic estimates could be clarified more explicitly. Overall, I see this as a solid and relevant workshop contribution, but not yet a clear accept.

---

### Official Review · Reviewer_2YnU · 2026-05-20
**This reviewer likes the general idea of a prior fitted network for the perturbation task the presentation and results are lacking. There is not sufficient description of the model to evaluate its novelty and results are lackluster. Interpretability of the model is given as this work's contribution but it is not evaluated on real world data to properly justify this claim.**

**Rating:** 5
**Confidence:** 4

**Review:**

## Summary:
&emsp;Authors present an interesting approach to the perturbation modelling task which incorporates naturally interpretable elements which could allow for a unified prediction interpretation approach. This approach uses purely synthetic examples to train a model for perturbation prediction tasks (target identification, structure, discovery, and perturbation effect prediction). PerturbPFN is competitive for effect prediction, best at structure discovery (within distribution task), and middling at target identification.
## Strengths:
&emsp;The idea of using a prior-fitted network and SCMs for the perturbation modeling task is interesting and worthy of exploration. Specifically the incorporation of an interpretable network prediction could have far reaching impacts for drug development and evaluation.
## Areas Of Improvement:
&emsp;Authors mention multiple times how the mechanistic approach to perturbation modelling has an interpretability benefit. They do not however, actively use their SCM to evaluate its interpretability apart from synthetic examples which are insufficient in the biological context to make any strong claims about interpretability. Authors should consider real world GRNs which may be inferrable to evaluate interpretability. E. Coli offers a potential context in which the GRNs are well understood and data is readily available. https://www.biorxiv.org/content/10.1101/2021.04.08.439047v2 offers a potentially useful resource to this end.
&emsp;Technical description of the model is lacking. I would want more explicit description of the networks and their construction in the format of a LaTeX algorithm with input, output and function definitions. As it currently stands I cannot garner a full understanding of the model definition from the text. A specific example of this are the “descriptor-to-intervention maps” in the latent SCM, which are mentioned but never described.
&emsp;The evaluation of models is presented in a way that is difficult to compare across works. Specifically the min-max normalization makes it difficult to directly compare performance. Authors should include the raw performance values in the supplemental. Additionally since you are predicting a distribution of outcomes, evaluation of the uncertainty in the predictive distribution would help to differentiate your approach and its abilities.
## Detailed Comments:
&emsp;Use of the word “atomic” in this context can be confusing as it seems to refer to individual graph elements. However one could reasonably be confused into thinking it refers to the chemical intervention interacting with actual molecular elements. This language should be clarified to remove ambiguity.
&emsp;In table 1 i am assuming bold is indicative of best. You bold the entire rightmost column and not individual elements. Similarly the other two tables have no bolded elements, this should be made consistent.
You mention multiple approaches for perturbation prediction (black-box, foundation models, and mechanistic). You however do not compare or discuss relative performance, it would be helpful for the broader community to discuss how your approach compares to those in other paradigms.
## Justification for Score:
&emsp;While there may be benefits to the approach of a PFN strategy to perturbation modelling there is insufficient evaluation to determine merits quantitatively. Additionally the explicit inclusion of a network in the modelling process without comparison to well understood gene regulatory networks makes it impossible to determine its interpretability and biological relevance.